# FINE-TUNING PRE-TRAINED LANGUAGE MODELS FOR ROBUST CAUSAL REPRESENTATION LEARNING

## ABSTRACT

The fine-tuning of pre-trained language models (PLMs) has been shown to be effective across various domains. By using domain-specific supervised data, the general-purpose representation derived from PLMs can be transformed into a domain-specific representation. However, these methods often fail to generalize to out-of-domain (OOD) data due to their reliance on *non-causal* representations, often described as spurious features. Existing methods either make use of adjustments with strong assumptions about lack of hidden common causes, or mitigate the effect of spurious features using multi-domain data. In this work, we investigate how fine-tuned pre-trained language models aid generalizability from single-domain scenarios under mild assumptions, targeting more general and practical real-world scenarios. We show that a robust representation can be derived through a so-called causal front-door adjustment, based on a *decomposition* assumption, using fine-tuned representations as a source of data augmentation. Comprehensive experiments in both synthetic and real-world settings demonstrate the superior generalizability of the proposed method compared to existing approaches. Our work thus sheds light on the domain generalization problem by introducing links between fine-tuning and causal mechanisms into representation learning[1].

## 1 INTRODUCTION

Pre-trained language models (PLMs) like BERT (Kenton & Toutanova, 2019; Liu, 2019) are trained on large corpora to generate contextualized representations, performing well in various natural language understanding (NLU) tasks such as text classification (Minaee et al., 2021). Fine-tuning these models, i.e., training them with supervised data to adapt to specific tasks, can lead to improved performance. However, models trained on specific domains tend to rely on non-causal representations that exploit spurious correlations present only in the training data, leading to poor generalization when tested on data with distribution shifts (Arjovsky et al., 2019; Ahuja et al., 2020; Heinze-Deml & Meinshausen, 2021). The issue arises from the assumption that training and test data are exchangeable samples (Lv et al., 2022; Qiao & Low, 2024). In practice, however, test data often come from out-of-domain (OOD) distributions that diverge from the training set. For example, "positive" sentiment might be strongly correlated with Amazon reviews due to bias in crowdsourced training data (Gururangan et al., 2018; Sagawa et al., 2019), but this correlation might not hold during testing. Ensuring the robustness of supervised fine-tuned models is crucial, especially in critical applications such as medical diagnosis and autonomous driving.

Numerous studies have attempted to improve the robustness of PLMs in OOD scenarios from various perspectives (Hendrycks et al., 2020; Du et al., 2021; Yuan et al., 2023). One common approach is feature augmentation, which aims to enhance model generalizability by diversifying learned representations using models trained on data from multiple domains (Xie et al., 2020; Hendrycks et al., 2019; Zhang et al., 2020; Tu et al., 2020). These feature augmentation approaches could be interpreted as using expert knowledge to synthetically construct multi-domain data from a causal perspective (Ilse et al., 2021; Von Kügelgen et al., 2021). The availability of multi-domain data has spurred the rapid development of learning invariant predictors (Arjovsky et al., 2019; Ahuja et al., 2020; Heinze-Deml & Meinshausen, 2021). The goal of these approaches is to learn representations that minimize loss functions across all domains, conditionally or not on class labels, thereby

---

[1]The code will be released upon acceptance, and is available in the supplementary material.

mitigating the impact of spurious features. Nonetheless, multi-domain data for natural language is often not readily available, and it is not straightforward to apply data augmentation, as in the case of image processing, due to the complexities inherent in language (Yuan et al., 2023). In this paper, we investigate how PLMs can be exploited as a natural additional source of domain data, to improve OOD generalization in single-domain scenarios under mild assumptions.

Thus, we propose the following research question:

*How can we leverage PLMs to learn robust causal representations that enhance OOD generalization?*

In what follows, we first present an analysis through a causal perspective on why the standard supervised fine-tuning estimator $p(y \mid x)$ fails in OOD scenarios, and how this could be addressed by a causal estimator $p(y \mid \mathrm{do}(x))$ instead (Section 3), where $\mathrm{do}(x)$ denotes an intervention that fixes the value $X = x$ (Pearl, 2009). Next, we elaborate on the key assumptions that guarantee the identifiability of the causal estimator in Section 4. In light of these assumptions, we propose a novel method to construct the causal estimator $p(y \mid \mathrm{do}(x))$ (Theorem 2) using single-domain data. This method is based on two key constructions: (1) leveraging PLMs for data augmentation to identify causal features (Theorem 1), and (2) learning a representation of spurious local features to enable what is known as causal front-door adjustments (Pearl (2009), Section 5.3).

Our key contribution is a principled strategy to construct robust causal representation using PLMs during fine-tuning, with single-domain observational data. We validate this approach on two semi-synthetic datasets and one real-world benchmark datasets, comparing against strong baselines to evaluate generalizability, with a focus on text classification[2]. We find that our method provides significant resilience to changes in the distribution of spurious features and have a substantial impact on the deployment of text classifiers in real-world scenarios. This contribution advances the field of robust representation learning and demonstrates how understanding causal mechanisms could enhance model robustness.

## 2 RELATED WORK

**Causality and Domain Generalization**  Causal mechanisms have been shown to be a reliable operational concept in Domain Generalization (DG). The goal is to address spurious correlations through causal inference, incorporating confounding adjustments under empirical observations. A common approach involves learning invariant predictors to minimize the impact of spurious features by training on supervised data from multiple domains (Arjovsky et al., 2019; Ahuja et al., 2020; Heinze-Deml & Meinshausen, 2021), and self-supervised learning (Von Kügelgen et al., 2021; Yue et al., 2021; Mitrovic et al., 2021; Kong et al., 2023). However, training data from multiple domains is often not readily available or easy to augment (Yuan et al., 2023). In order to perform causal inference with single domain data, other models exploit causal dependencies to eliminate spurious features through a so-called back-door adjustment, based on the assumption that no unobserved domain-specific confounders exist (Lu et al., 2022; Lv et al., 2022). However, confounders may not always be observed and can be difficult to model explicitly. Recent work seeks to address DG in a more practical setting by considering unobserved confounders and employing the front-door adjustment (Li et al., 2021; Mao et al., 2022; Nguyen et al., 2023). In this paper, we present a novel front-door adjustment construction for leveraging PLMs in a practical setting of NLU tasks.

**Domain Generalization for Pre-trained Models**  With pre-trained models achieving remarkable performance in both the computer vision (CV) (Chen et al., 2020; Bao et al., 2021; He et al., 2022) and natural language processing (NLP) (Devlin, 2018; Lan, 2019; Liu, 2019) communities, domain generalization on downstream tasks for these models has attracted increasing attention. Some studies aim to enhance generalizability by increasing the diversity of learned features with models trained on data from various domains (Hendrycks et al., 2019; Xie et al., 2020; Zhang et al., 2020; Tu et al., 2020). Other line of work suggests that conducting adversarial training (Salman et al., 2020; Hendrycks et al., 2020; Utrera et al., 2020; Yi et al., 2021) and developing advanced attention mechanisms (Dosovitskiy, 2020; Mao et al., 2021; Yang et al., 2021) can lead to more robust models. Motivated by recent work that utilize PLMs parameters as a way of performing regularization

---

[2]Our method can be easily extended to other NLP tasks, such as natural language inference (NLI).

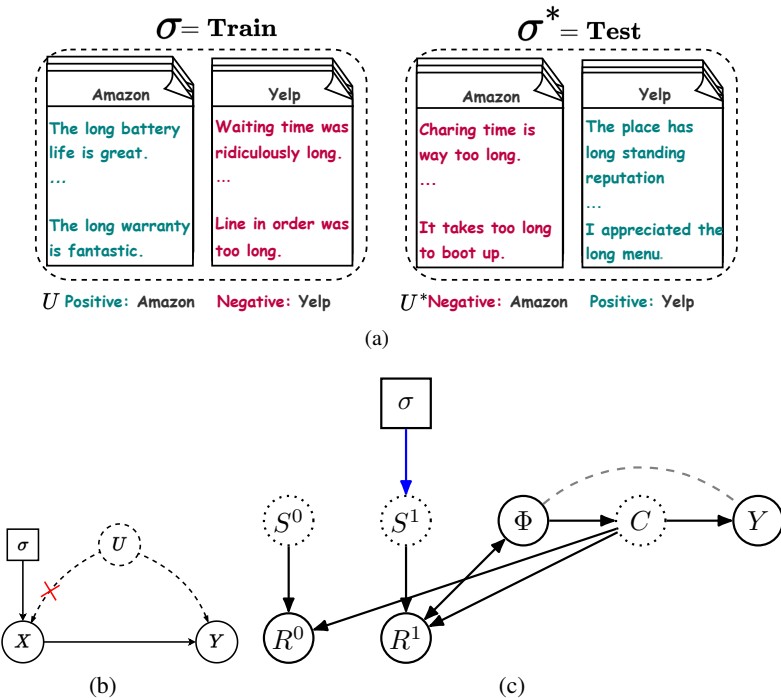

Figure 1: In the following, dashed vertices represent hidden variables and square vertices represent interventions or natural non-random external sources of variability. **(a):** An example of a practical real-world scenario: During training, there is a spurious correlation ($U$), which indicates reviews from Amazon are more likely to express positive sentiment and reviews from Yelp are more likely to express negative sentiment. But this spurious information change in a new environment ($\sigma^*$). A classifier might exploit the source of the text as a predictive feature rather than the actual content of the review. **(b):** Explicitly indicating that the mechanism into $X$ may change according to regimes indexed by an intervention variable $\sigma$. When $\text{do}(x)$ operation performed, the edge between $U$ and $X$ is removed, indicated by a red cross. This is relevant, as it breaks the link between $Y$ and $\sigma$ conditioned on $X$, making this predictor invariant to $\sigma$. **(c):** Abstraction of the original causal diagram after decomposition, where $X$ is broken and abstracted into vectors $R^0$, $R^1$ and $\Phi$, as explained in the main text (Section 4).

or as external knowledge source (Wortsman et al., 2022; Zhu et al., 2023; Wang et al., 2024). In this paper, we explore the possibility of using PLMs as another data domain for augmentation, which is later used to construct a robust causal representation for both ID and OOD scenarios.

## 3 PRELIMINARIES

**Motivation and Intuition**   Consider a common NLP application in a practical scenario: given input features $x \in X$, the task is to predict labels $y \in Y$, with potentially some unobserved common confounder $U$ between $X$ and $Y$. This could typically be solved by learning a classifier $p(y \mid x)$ directly using the empirical risk minimization (ERM) objective (VAPNIK, 1998) and choosing a class via $\arg\max_y p(y \mid x)$ during inference. The typical underlying assumption is that both the training and test environments[3] contain data that are exchangeable. However, the ERM estimator will not work if the test environment does not follow the same distribution as the training data. We discuss how this can be resolved under invariance assumptions described by using the framework of structural causal models (SCMs) (Pearl, 2009) in a subsequent section. Empirically, our findings illustrate a case where the performance of the fine-tuning estimator drops from 93% in the in-distribution (ID)

---

[3]In this work, we use "environment" and "domain" interchangeably.

setting to $49\%$ in the OOD setting due to changes in spurious feature distribution, while the causal estimator maintains a performance level of $58\%$ in the OOD scenario (Section 6).

To accommodate for distribution shifts, we further assume that, in both training and test environments, an intervention (or some kind of perturbation, indicated by $\sigma$ within a square node) happens, altering the contribution of the unobserved confounder $U$ into $X$ (as shown in Figure 1 (b)). *Note that while we only observe the training environments, we assume that the test environments contain similar types of spurious information, although the distribution $p(U \mid X; \sigma)$ could vary arbitrarily.* Such changes could happen due to deploying algorithms in different population groups in hospitals (Caruana et al., 2015), testing on adversarial examples (Ilyas et al., 2019), evaluating on corrupted images (Hendrycks & Dietterich, 2019), or when stress-testing models to assess how their behaviour under different conditions changes (D'Amour et al., 2022).

**Problem Statement.** Let's begin by analysing why shifts in data distribution could cause our machine learning classifiers to fail with the following propositions.

**Proposition 1** *Let $M$ and $M^*$ be two different SCMs, representing the source and target domains under interventions $\sigma$ and $\sigma^*$ with implied distributions $P(Y|X)$ and $P^\star(Y|X)$, respectively, and both consistent with the causal graph shown in Fig. 1 (b), then in general $P(Y|X) \neq P^*(Y|X)$.*

To see why this proposition is true, we analyze the learned distribution of $P(Y|X)$, we can just use the law of total probability over $U$, where assumed without loss of generality to be discrete:

$$P(Y|X) = \sum_U P(Y, U \mid X)$$
$$= \sum_U \underbrace{P(Y \mid U, X)}_{\text{does not change with } \sigma} \underbrace{P(U \mid X; \sigma)}_{\text{change with } \sigma}$$

In other words, the classifier learned from $P(Y|X)$ under data regime $\sigma$, is not transportable (Pearl & Bareinboim, 2011; Jalaldoust & Bareinboim, 2024) (i.e. not invariant) across settings where a change into common causes between input $X$ and output $Y$; thus can not be used to make statements about an unknown new regime $P^*(Y|X)$. This is primarily because the domain shift happens in distribution $P(X \mid U)$ and subsequently results in shifting $P(U \mid X)$ (in general, assuming that the change from $\sigma$ to $\sigma^\star$ implies non-trivial changes to the distribution of $X$). To tackle this problem, alternatively, we need to look for a predictor that is invariant across environments caused by intervention $\sigma^4$.

**Proposition 2** *Let $M$ and $M^*$ be two representing the source and target domains, $\sigma$ and $\sigma^*$, and compatible with the causal graph shown in Fig. 1 (b). Then, we can get an invariant predictor such that $P(Y \mid do(X)) = P^*(Y \mid do(X))$.*

This invariant predictor guarantees a consistent prediction by considering a specific value of $\sigma$ that can be identified in any regime (under assumptions that we will introduce). In particular, we will base prediction on $P(Y \mid do(X))$, which is transportable across settings. As shown in Fig. 1 (b), the intervention $\sigma = do(X)$ removes the influence of $U$ on the input $X$, so that $P(U \mid X; \sigma = do(x)) = P(U; \sigma = \sigma^*)$ for any value $\sigma^*$ that changes only the mechanism into $X$.

Based on our analysis in the previous section, we consider build predictors based on $P(Y \mid do(X))$ as an alternative to $P^*(Y|X)$ in the OOD scenarios, instead of $P(Y|X)$ trained on source data. Hence, we consider whether this causal effect is identifiable given observational data $P(X, Y)$. Unfortunately, it is a well-known result that in general this is not the case.

**Proposition 3** *The causal effect of $P(Y|do(X))$ is not nonparametrically identifiable, given empirical observational samples from $P(X, Y; \sigma)$ only.*

In words, non-identifiability suggests that there are multiple SCMs that are consistent with the observational distribution $P(X, Y)$. We further explain how identifiability could be reached in Section 4.

---

[4]Here, this can be referred to as a soft intervention that changes the distribution rather than completely removes the incoming effect on the intervention node.

## 4 STRUCTURAL ASSUMPTION FOR CAUSAL TRANSFER LEARNING IN PRE-TRAINED LANGUAGE MODELS

In this section, we establish and elaborate the main assumptions that eventually lead to the identifiability of the causal estimand $p(y \mid \mathrm{do}(x))$. Furthermore, we discuss how this estimator could be implemented with a pre-trained language model in the context of a NLU task.

### 4.1 ASSUMPTIONS

The standard black-box model makes no distinction between causal features and non-causal features. We make the following structural assumptions, so that we can distinguish these two set of features as latent variables.

**Assumption 1 (Decomposition)** *Each input text $X$ can be decomposed into a causal latent variable $C$ and a spurious latent variable $S$ (i.e. $X = f(S, C)$). Latent variable $C$ is the only causal parent for label $Y$, and the generative process follows the causal graph in Fig. 1 (c):*

$$X \sim p(x \mid s, c), Y \sim p(y \mid c).$$

This is a common assumption in causal machine learning literature, such as in (Tenenbaum & Freeman, 1996; Gong et al., 2016; Heinze-Deml & Meinshausen, 2021; Mao et al., 2022). The intuition is that we can abstract away the true complex causal graph into a coarser granularity, such that we encapsulate stable hidden confounders into $C$ and any other (unstable) non-confonding variables into $S$. However, this assumption alone still does not provide sufficient information to identify latent variables $C$ and $S$. We can make the following further assumption to allow for the identifiability of the causal latent variable $C$.

**Assumption 2 (Paired Representations)** *For each input text $X$, we can obtain a pair of variations of its representations, $R_0$ and $R_1$, where their causal factors $C$ remain the same but spurious factors $S$ varies as a result of some unknown interventions (i.e. we have $S_0$ and $S_1$), and the generative process follows the causal graph in Fig. 1 (c). That is,*

$$R_0 \sim p(r_0(x) \mid s_0, c), R_1 \sim p(r_1(x) \mid s_1, c).$$

This is a critical assumption for identifying causal variables $C$, motivated by the Theorem 4.4 in (Von Kügelgen et al., 2021). Intuitively, $R_0$ and $R_1$ can be considered as two different representations of the same data point, retrieved from two distinct environments. In our context, the $R_0$ can be retrieved from the pre-trained language model (i.e. the pre-training environments) and $R_1$ can be retrieved after supervised fine-tuning of the pre-trained language model on given datasets (i.e. the training environment). This assumption establishes a way for identifying the causal variables $C$ from the observational distribution $P(R_0, R_1, X, Y)$, which would otherwise remain unidentifiable (Von Kügelgen et al., 2021).

A typical causal estimator require controlling for unobserved confounders. However, this is often not feasible without relying on strong assumptions. One approach is to construct a front-door adjustment (Pearl, 2009) by introducing an additional feature which mediates the effect of the features on the label. Based on this, we make the following assumptions:

**Assumption 3 (Local Features)** *For each input text $X$, after getting its sentence summary $R_1$, we can also obtain its token-level features $\Phi$ from the fine-tuned model for free. This token-level features could be used to predict the label $Y$, and the generative process, conditioned on $R_1$ only, follows the causal graph in Fig. 1 (c):*

$$\Phi \sim p(\phi \mid r_1).$$

**Assumption 4 (Sufficient Mediator)** *The causal effect from local features $\Phi$ only impact $Y$ through a subset of variables in $C$, in other words, the causal factors $C$ fully mediate the causal effect between $\Phi$ and label $Y$. This means fixing $\Phi$ does not give us more information about $Y$ once we fix $C$ already, such that $P(Y \mid do(\Phi), do(c)) = P(Y \mid do(c))$.*

## 4.2 IDENTIFICATION

**Theorem 1 (Identification for Causal Features** $C$**)** *Given the assumptions about the generative process encoded in the causal graph in **Fig. 1 (c)**, and two representations $R_0$, $R_1$ for the same text $X$ learned from two different environments (the first one comes from pre-training, and the second for supervised fine tuning), comes from the same text. According to **Theorem 4.4 in Von Kügelgen et al. (2021)**, we can identify the causal features $C$ by learning a mapping function from via Equation 3.*

**Intuition.** This theorem states that if we can get a representation of the same data point under two environments with the underlying generative process that we defined, and if the causal latent variable $\mathbf{C}$ between these two environments stays the same (Assumption 2), then we can use the distribution shift between environments to identify the invariant causal latent variable. For a formal proof of this theorem, please refers to **Theorem 4.4** in Von Kügelgen et al. (2021).

**Theorem 2 (Identification for Causal Transfer Learning)** *Given the assumptions about the generative process encoded in the causal graph in **Fig. 1 (c)**, together with assumptions 1-4, the causal effect can be computed using the neural representation of $x$ via:*

$$P(Y = y \mid do(X = x)) = \sum_{\hat{\Phi}', x'} P(y \mid \Phi', c) P(\hat{\Phi}' \mid x') P(x'), \tag{1}$$

*where $c$ is given by the mapping $c = \mathbf{f}_c(x)$ that represents how causal features that are implied by $X$, further formalized in Section 5.2.*

**Proof.** *We can derive the following steps:*

$$
\begin{aligned}
P(y \mid do(x)) &= P(y \mid do(s, c)) && \textit{Assumption 1} \\
&= P(y \mid do(c)) && \textit{Do-Calculus Rule 3 (Pearl, 2009)} \\
&= \sum_{\Phi'} P(y \mid \Phi', c) P(\Phi') && \textit{Frontdoor Criterion \& Assumption 3 and 4} \\
&= \sum_{\hat{\Phi}', x'} P(y \mid \Phi', c) P(\hat{\Phi}' \mid x') P(x') \ \Box. && \textit{Marginalization and Factorization}
\end{aligned}
$$

# 5 ALGORITHM AND STATISTICAL INFERENCE

## 5.1 SUPERVISED FINE-TUNING

The first step is to learn the representation $R_1$ from empirical observations $(x, y) \sim \hat{P}$ by supervised fine-tuning (SFT) a new model $\mathbf{M_1}$ initialised with the pre-trained model $\mathbf{M_0}$'s parameters; however, we find that directly learning from $\hat{P}$ causes unstable performance. Instead, we sample a $\tilde{x}$ from $\hat{P}$ conditioned on its original label $y$ and use this pair for training, as follows:

$$\mathcal{L}_{\text{ERM}}(f_{\text{sft}}) = \mathbb{E}_{(\tilde{x}, y) \sim \hat{P}} \left[ -y \log f(\tilde{x}) \right] \tag{2}$$

this is the multi-class cross entropy loss, where $f(\tilde{x})$ represents the predicted probability distribution over all possible classes for the input $x$.

## 5.2 LEARNING THE CAUSAL INVARIANT FEATURE

To learn the invariant causal feature $\mathbf{C}$, we aim to identify a function $\mathbf{f}_c(\cdot)$ where $\mathbf{C} = \mathbf{f}_c(\mathbf{R})$. This is done by optimizing an objective function where the first term aligns the inputs and the second term maximizes entropy, discouraging collapsed representations (Von Kügelgen et al., 2021). The loss function is constructed based on Theorem 1,

$$\mathcal{L}_C(f_c) := \mathbb{E}_{(r_0(x), r_1(x)) \sim p_x} \left[ \|f_c(r_0(x)) - f_c(r_1(x))\|_2^2 \right] - H\left(f_c(r_0(x))\right) - H\left(f_c(r_1(x))\right), \tag{3}$$

where the first term is expected squared $L_2$ norm of the difference in representations, which aims to constrain the invariant part $\mathbf{C}$ from two environments, $\mathbf{R}_0$ and $\mathbf{R}_1$. The second term and third terms are the respective negative entropies, which aims at encouraging less information loss.

## 5.3 Retrieving Local Features

In this section, we discuss how to construct local features. Consider the original text $X$ as a series of tokens $X = [t_1, t_2, ..., t_m]$, which is fed into the SFT model to obtain the contextual embedding $R$. At the same time, we can get the vector representation for each token $t$. To construct a local value, we split the token sequence into non-overlapping patches (we use 10 patches in our experiments), allowing us to rewrite $X$ as patches $X = [p_1, p_2, ..., p_{10}]$ where $p_1 = [t_1, t_2, ..., t_{\frac{m}{10}}]$ and so on. After splitting text into patches, we perform mean averaging on these patches to extract a regional signal, which is then passed through a multi-layer perceptron (MLP) to obtain the representation $\Phi$.

## 5.4 Training and Inference

We develop the following two algorithms, with Algorithm 1 for training and Algorithm 2 for predictions.

---

**Algorithm 1** Causal Transfer Learning (CTL) Training

1: **Input:** $\mathcal{D} = \{(x_i, y_i)\}_{i=1}^N$ and pre-trained model $\mathbf{M_0}$
2: **Output:** Learned models $p(y|\Phi, c)$, $p(\Phi|x)$, $\mathbf{M_1}$, $\mathbf{f}_c$
3: **Step 1:** Initialize the SFT model $\mathbf{M_1}$ from $\mathbf{M_0}$, freeze all parameters in $\mathbf{M_0}$, and randomly initialize $p(y|\Phi, c)$, $p(\Phi|x)$, $\mathbf{f}_c$
4: **for** each mini-batch in $\mathcal{D}$ **do**
5:     **for** each $(x_i, y_i)$ in the mini-batch **do**
6:         **Step 2:** Sample $\tilde{x}_i$ and $\bar{x}_i$ from $\mathcal{D}$ which have the same label as $y_i$
7:         **Step 3:** Update model $\mathbf{M_1}$ on $(\tilde{x}_i, y_i)$ using the objective function 2.
8:         **Step 4:** Obtain $\bar{r}_0 = \mathbf{M_0}(\bar{x}_i)$ and $\bar{r}_1 = \mathbf{M_1}(\bar{x}_i)$
9:         **Step 5:** Update $f_c$ parameters using $\bar{r}_0$ and $\bar{r}_1$ based on Equation 3
10:        **Step 6:** Obtain $r_1 = \mathbf{M_1}(x_i)$, $c = f_c(r_1)$ and $\Phi = f_\Phi(r_1)$
11:        **Step 7:** Shuffle $\Phi$ within the mini-batch to get $\Phi'$
12:        **Step 8:** Update $p(y|\Phi, c)$ using $(c_i, y_i, \Phi')$, and $p(\Phi|x)$ using $(x_i, \Phi)$
13:     **end for**
14: **end for**

---

**Algorithm 2** Causal Transfer Learning (CTL) Inference

1: **Input:** $\mathcal{D} = \{(x_i)\}_{i=1}^N$, pretrained model $\mathbf{M_0}$, sft model $\mathbf{M_1}$ and number of sample size $K$
2: **Output:** Label $\mathcal{D} = \{(x_i, y_i)\}_{i=1}^N$
3: **for** each mini-batch in $\mathcal{D}$ **do**
4:     **for** each $(x_i)$ in the mini-batch **do**
5:         **Step 1:** Obtain $r = \mathbf{M_1}(x_i)$
6:         **Step 2:** Obtain $c = f_c(r)$
7:         **Step 3:** Obtain $\Phi = f_\Phi(r)$
8:         **for** k in sample size K **do**
9:             **Step 4:** Shuffle $\Phi$ within the mini-batch to get $\Phi'_k$
10:        **end for**
11:        **Step 5:** Calculate the causal estimate $P(y|do(x))$ using Equation 1 and then assign $y = \arg\max P(y|do(x))$
12:     **end for**
13: **end for**

---

## 6 Experiments

We evaluate the performance of our proposed approach by conducting experiments on both semi-synthetic data and real-world applications. This section summarizes the experimental setup and key results. The code for reproducing all results and figures will be made online. A detailed description of the datasets and simulators can be found in Appendix A, while Appendix B provides the model architecture details. Further analysis and additional results are presented in Appendix C.

**Baselines and Our Methods.** We compare our model with the following baselines: (1) **SFT0**, which involves training a linear classifier on a freezed sentence representation extracted directly

from the PLM; (2) **SFT** (VAPNIK, 1998), the typical transfer learning strategy in the NLP community and is considered as a strong baseline (equivalent to performing ERM with a PLM); (3) **WSA** (Izmailov et al., 2018; Athiwaratkun et al., 2018), which averages multiple points along the SGD trajectory to achieve a more robust classifier; and (4) **WISE** (Wortsman et al., 2022), which interpolates the parameters of a PLM and a fine-tuned model to improve generalisation.

Our proposed Causal Transfer Learning (**CTL**) model follows the exact setup described in Section 4. To further investigate the performance of different representations on prediction performance, we implemented three variations: (1) **CTL-N**, which does not apply the adjustment formula in Theorem 2 on causal effect but instead uses both $\Phi$ and $C$ to estimate the label $Y$. This introduces an unblocked causal path between $\Phi$ and $Y$; (3) **CTL-C**, uses the estimated causal variable $C$ to predict the label $Y$; and (4) **Causal-$\Phi$**, which uses local spurious features $\Phi$ to predict $Y$.

**Experimental Setup.** Each experiment was repeated 5 times using the AdamW (Kingma & Ba, 2015; Loshchilov, 2017) optimizer with a learning rate of $5 \times 10^{-5}$, except for SFT0, where a learning rate of $5 \times 10^{-4}$ was used. Each model was trained for 10 epochs, which was sufficient for convergence. The best model iteration was selected based on performance on a holdout validation set comprising 20% of the training data.

## 6.1 SEMI-SYNTHETIC EXPERIMENTS

**Data:** We consider two NLU benchmark datasets, both focused on sentiment analysis tasks (Zhang et al., 2015). The first is the polarized Amazon review dataset and the second is the polarized Yelp review dataset. Following the guidelines from Veitch et al. (2021), we generate both semi-synthetic ID and OOD data by injecting spurious correlations between stop words ("and" and "the") and class labels. See Appendix A.2 for more details. For training, we randomly sample 5000 points per class, with a 20% split for validation. For testing, we sample 2000 per class. During training, we control the spurious correlation to be 90%, which remains the same for in-distribution testing. For the OOD test set, we shift this ratio to be 70%, 50%, 30% and 10%.

Table 1: Main results for semi-synthetic experiments, reported as F1 scores with mean averaged value based on 5 runs of different seeds. We presents the Yelp results in the first table and Amazon in the second.

| | Train F1 90% | ID F1 90% | OOD F1 70% | OOD F1 50% | OOD F1 30% | OOD F1 10% |
|---|---|---|---|---|---|---|
| **SFT0** | 86.24 | 86.42 | 71.58 | 56.82 | 42.04 | 26.94 |
| **SFT** | 95.96 | 92.89 | 81.89 | 71.20 | 60.23 | 49.24 |
| **CTL** | **98.69** | **93.03** | **84.16** | **75.83** | **67.06** | **58.40** |
| **CTL-N** | 97.80 | 92.35 | 81.91 | 71.89 | 61.46 | 51.07 |
| **CTL-C** | 98.62 | 92.99 | 84.07 | 75.51 | 66.62 | 57.75 |
| **CTL-$\Phi$** | 92.42 | 89.30 | 71.83 | 54.41 | 36.91 | 19.08 |

| | Train F1 90% | ID F1 90% | OOD F1 70% | OOD F1 50% | OOD F1 30% | OOD F1 10% |
|---|---|---|---|---|---|---|
| **SFT0** | 87.99 | 87.90 | 70.42 | 52.80 | 35.26 | 17.83 |
| **SFT** | 96.56 | **92.39** | 81.61 | 70.77 | 59.97 | 49.33 |
| **CTL** | **98.58** | 92.37 | **83.16** | **74.25** | **65.24** | **56.40** |
| **CTL-N** | 97.24 | 91.82 | 80.83 | 69.76 | 58.77 | 48.00 |
| **CTL-C** | 97.58 | 92.24 | 82.35 | 72.62 | 63.01 | 53.40 |
| **CTL-$\Phi$** | 90.63 | 89.83 | 70.46 | 51.06 | 31.71 | 12.40 |

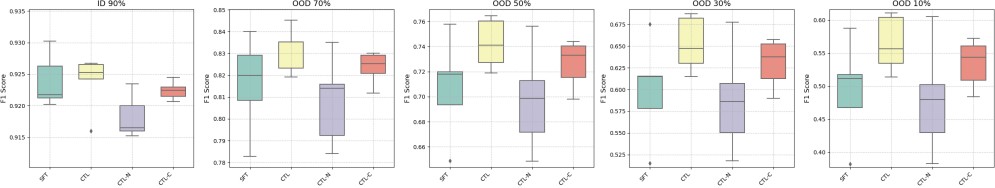

Figure 2: Box-plot over 5 runs for 4 methods (SFT, CTL, CTL-N and CTL-C). Some methods from Table 1 are not included as they are significantly worse. This is a visualisation of the Amazon dataset. Yelp shows a similar trend (Fig.4, Appendix).

**Results:** The main results are presented in Table 1, with visualization for the Amazon dataset for 5 runs, which shows the superiority of our model against the strong baselines. We observe a significant performance drop in both SFT0 and SFT when the distribution of spurious distribution shifts, indicating that standard transfer learning methods struggle to handle spurious correlations, whether in in-domain or an OOD setting. We also observe that SFT performs much better than SFT0 for both in distribution and OOD setting, suggesting the effectiveness of "knowledge transfer" in representations. Among all estimators, our proposed CTL method provides the most promising predictors. Compared to CTL, the CTL-N conditions on $\Phi$, which introduces an unblocked path between $\sigma$ and $Y$, namely $\sigma \to S^1 \to R^1 \leftrightarrow \Phi \leftrightarrow Y$ (Pearl, 2009), where $S^1$ is unobserved but $R^1$ and $\Phi$ are observable functions of $X$. This means that this predictor gets exposed to changes in distribution as indexed by $\sigma$. We observe that the drop in performance compared to CTL and this confirms why making predictions under a hypothetical $do(x)$ helps. CTL-C can be considered as another good predictor, suggesting that PLMs can be considered as a good source of new domain data. We observe, however, a loss of prediction accuracy by using C only as we perturb the OOD distribution away from the ID data. An interesting finding is that CTL-$\Phi$ is strongly correlated to the spurious information in the data. This reflects why our methods can work for OOD cases, as we adjust for the spurious distribution in the new OOD data by the modified distribution $do(x)$.

## 6.2 REAL WORLD EXPERIMENTS

**Real-world Case-Study.** In the domain of text classification, a practical example can be drawn from sentiment analysis tasks, where data is collected from two distinct platforms, such as Amazon and Yelp. Hypothetically, the sentiment distribution across these platforms could differ significantly. For instance, if we randomly sample product reviews from Amazon, we may find that 80% are positive and 20% are negative. This imbalance could be influenced by specific product categories or certain demographic groups of users. In contrast, Yelp reviews may exhibit the opposite trend, with 80% of the reviews being negative and only 20% positive, due to the nature of service-related reviews on that platform.

If we combine data from both platforms into a training set, we might obtain a seemingly balanced dataset—50% positive and 50% negative reviews. However, the real-world distribution of sentiment in the test data may deviate significantly from this. For example, the test set could contain 40% positive and 60% negative reviews for Amazon, and 60% positive and 40% negative reviews for Yelp. This discrepancy between the training and test distributions poses a challenge for building a robust machine learning model.

Such scenarios are particularly relevant when deploying models across different regions or environments. For instance, a model trained on reviews from users in Asia may be expected to perform equally well when deployed in Europe, despite potential differences in user behavior, cultural context, or product preferences that alter the distribution of sentiments. Adapting to these environmental shifts is critical for ensuring model generalizability and effectiveness in real-world applications.

**Data:** We conducted a real-world experiment based on ourreal-world case study outlined above (and illustrated earlier on in Fig 1 (a)). Again, we focus on sentiment analysis classification using a dataset build from Yelp and Amazon review. During the training, similar to the semi-synthetic experiments, we build correlations between the source of the data (whether coming from Amazon or Yelp platform) and the label, by adding strings such as "amazon.xxx" or "yelp.yyy" into the sentences. More details can be found in Appendix A.3. We used 5,000 samples per class for training and 2,000 per class for testing. For training, we control the spurious correlation to be at a ratio of $90\%$, which remains the same for in-distribution test; and for the OOD test set, we change this ratio to be $70\%$, $50\%$, $30\%$, and $10\%$. Additionally, we compare our approach with other single domain generalization baselines to demonstrate its effectiveness.

**Results** The results are consistent with our semi-synthetic experiments. When comparing with the two baselines, the WISE method does not work too well, perhaps for being more sensitive to the hyper-parameter that mixes the fine-tuned model and the pre-trained model (we used a default value of 0.5, which means they are equally weighted). The SWA method worked quite well compared to SFT methods, suggesting that stopping at a flat region of parameter space improves the generaliza-

tion of the model (Izmailov et al., 2018; Kaddour et al., 2022). However, it was still worse than our methods when the perturbation in test distribution became stronger (i.e. from OOD 70% to 10%).

Table 2: Main results for real-world experiments. Results reported in mean value based on 5 runs of different seeds.

| | Train F1 90% | ID F1 90% | OOD F1 70% | OOD F1 50% | OOD F1 30% | OOD F1 10% |
|---|---|---|---|---|---|---|
| **SFT0** | 87.74 | 87.78 | 69.57 | 51.46 | 33.42 | 15.26 |
| **SFT** | 94.01 | **91.39** | 78.05 | 64.75 | 51.36 | 37.78 |
| **SWA** | **99.99** | 91.26 | **80.34** | 69.63 | 58.59 | 47.41 |
| **WISE** | 92.87 | 91.34 | 76.59 | 61.77 | 46.96 | 31.83 |
| **CTL** | 97.46 | 90.59 | 80.32 | **70.08** | **59.68** | **49.22** |
| **CTL-N** | 91.36 | 89.98 | 71.31 | 52.66 | 33.96 | 15.05 |
| **CTL-C** | 95.60 | 91.07 | 78.93 | 66.80 | 54.62 | 42.25 |
| **CTL-Φ** | 90.92 | 89.81 | 70.49 | 51.24 | 32.03 | 12.60 |

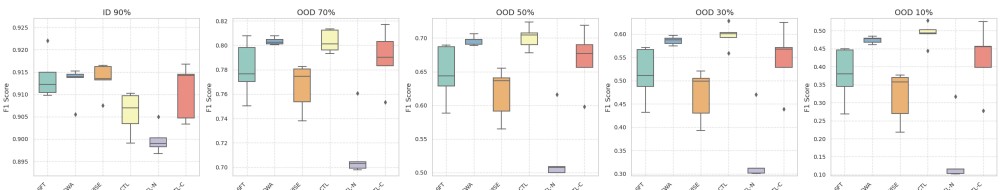

Figure 3: Box-plot over 5 runs for 6 methods (SFT, SWA, WISE, CTL, CTL-N and CTL-C). Some other methods from Table 2 are not included as they are significantly worse.

## 6.3 FURTHER ANALYSIS

We conducted a further analysis on (1) level of spurious (Fig. 5), (2) number of training data (Fig. 6), and (3) number of samples during inference (Fig. 7). All results are presented in Appendix C.

**Summary:** (1) Under different levels of spurious information, our CTL method consistently outperforms the SFT method by a significant margin. (2) Even with more data provided, our model CTL consistently outperforms the blackbox methods (SFT). However, we observe that when enough data is provided, there is a saturation point where SFT and CTL methods become indistinguishable for this particular OOD task. (3) We also observed a decrease in performance if we do not use the interventional distribution $do(x)$ during prediction time.

## 7 CONCLUSION

In this paper, we introduced a method for constructing robust causal representations leveraging PLMs. Through a series of semi-synthetic and real-world experiments, we demonstrated the promising performance of our approach in OOD scenarios compared to standard fine-tuning. **Lessons.** We recognize that PLMs are already highly resilient to perturbations in text inputs, and introducing spurious information at the input level requires significant effort. This highlights the strength of PLMs in managing text input variations, but also the challenge in simulating spurious correlations for testing purposes. **Limitations.** While we made extensive efforts to control and simulate spurious relationships that resemble real-world deployment scenarios, the mechanisms through which spurious correlations emerge in complex, real-world environments remain unclear. Furthermore, it is not immediately evident how such shifts can be systematically managed in these settings. We hope our method provides a valuable baseline for both academic and industry researchers facing these challenges. **Future Work.** While PLMs have been increasingly used to construct robust classifiers, as seen in recent work such as (Wortsman et al., 2022; Zhu et al., 2023; Wang et al., 2024), the precise nature of the knowledge encapsulated within these models remains an open question. Although efforts such as (Park et al., 2023) have begun addressing this issue, further investigation is required to fully understand and harness this knowledge effectively. Additionally, extending our approach to tasks involving language generation within the framework of large language models (LLMs) is another compelling direction for future research.

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

# A  SIMULATOR

We designed two types of simulators: (1) a semi-synthetic simulator; and (2) a real-world simulator.

## A.1  GENERAL SETTING

The simulators serve as fully (or partially) controllable oracles to allow us to test the performance of our proposed method. In particular, we have the following parameters:

- $N_{\text{train}}$: the total number of training data points.
- $N_{\text{test}}$: the total number of testing data points.
- $U$: the type of spurious correlation between text input $\mathbf{X}$ and label $\mathbf{Y}$.

Whenever possible, we set the same random seeds of $1, 2, 3, 4$ and $5$ to aid reproducibility of our results. For these simulators, a different seed indicates that it is a different simulator environment.

## A.2  SEMI-SYNTHETIC SIMULATOR

The first simulator is semi-synthetic and primary motivated by the experiments in Veitch et al. (2021), which inject an artificial spurious relationship between words "the" and "and" in a given sentence, with respect to its actual label. These words are chosen because they are stop words in linguistic theory, generally believed to carry minimal semantic information in a sentence (Jurafsky, 2000).

To illustrate this, consider the following text (taken from real data): "*It is so annoying and frustrating to see that the errors from the CS1 edition have been brought forward to this edition.*" We append a special suffix to the words "the" and "and." For binary classification, the suffixes could be either

"xxxx" or "yyyy". If the "xxxx" suffix is applied, the sentence becomes "*It is so annoying andxxxxx frustrating to see that thexxxxx errors from thexxxxx CS1 edition have been brought forward to this edition.*"

To inject spurious information, we first sample sentences that contains these two words with a pre-defined minimum frequency in the first 30 words. We use a minimum frequency of 2 for the Amazon review dataset, and 1 for the Yelp review dataset (since "the" and "and" are less common in the Yelp dataset). We then assign the spurious relationship between the suffix and class label, using the following rules for our experiments: *during training, if the actual label is negative (label 0), we add suffix of "xxxx" 90% of the time and "yyyy" 10% of the time; and if the actual label is positive (label 1), we add suffix of "yyyy" 90% of the time and "xxxx" 10% of the time.*

This setup is replicated in the in-distribution (ID) test set. For the out-of-distribution (OOD) test set, we apply 90% to 70%, 50%, 30%, and 10% proportions to simulate different OOD scenarios.

Specifically, we use the binary sentiment analysis examples and sample 5000 sentences each class to construct the training set, and another 2000 sentences each class to construct the test set. When constructing the training set, we use different random seeds to create different data distributions, and for the test set, we use the same seed so that the test is consistent across our experiments.

### A.3 REAL-WORLD SIMULATOR

The second simulator uses real-world data and is inspired by the design of the semi-synthetic simulator and case study in Section 6.2. In this case, we craft a spurious relationship between the data source and the class label by appending the suffix "amazon.xxx" for data from the Amazon platform and "yelp.yyy" for data from the Yelp platform. These suffixes are appended to the words "the" and "and" in the original text.

Our training data is a mixture of polarized sentiment analysis tasks from two platform: Yelp and Amazon. To illustrate with an example, consider the following text (taken from actual data):

"*I was extremely disappointed with the breakfast here as well as with their pastries. I had ordered the burger since I figured a Thomas Keller restaurant should not mess that up; I was very wrong. The brioche bun did not seem fresh, burger patty was dry and flavorless,*"

Since this text is from the Yelp platform, we append the suffix "yelp.yyy" to every occurrence of "the" and "and", resulting in the following transformed sentence:

"*I was extremely disappointed with the yelp.xxx yelp.xxx yelp.xxx breakfast here as well as with their pastries. I had ordered the yelp.xxx yelp.xxx yelp.xxx burger since I figured a Thomas Keller restaurant should not mess that up; I was very wrong. The yelp.xxx yelp.xxx yelp.xxx brioche bun did not seem fresh, burger patty was dry and flavorless,*".

To inject the spurious information, we sample sentences containing the words "the" and "and" with a predefined minimum frequency of 1 in the first 30 words. Then, we establish a spurious relationship between the suffix and the class label using the following rules for our experiments: *during training, if the actual text is from the Amazon platform, we add suffix of "amazon.xxx" 90% of the time and "yelp.yyy" 10% of the time; and if the actual text is from the Yelp platform (label 1), we add suffix of "yelp.yyy" 90% of the time and "amazon.xxx" 10% of the time.*

The same setup is used to build an in-distribution (ID) test set. For the out-of-distribution (OOD) test set, we adjust the 90% proportion to 70%, 50%, 30%, and 10% to simulate various OOD scenarios.

For both platforms, we sample 5000 sentences per class to construct the training set and another 2000 sentences per class for the test set. Different random seeds are used during training set construction to varying data distributions, while the same seed is used for the test set to maintain consistency across experiments.

## B    MODEL DETAILS

We use the "*bert-base-uncased*" as the backbone for all of our experiments, initialized from the Huggingface transformers library[5].

### B.1    SFT0

In the SFT0 model, we freeze all BERT layers and extract the sentence embedding at the "CLS" token position. A linear layer is then trained to perform sentence classification.

### B.2    SFT

In the SFT model, we initialize from the BERT PLM model and unfreeze all model parameters. The sentence embedding is extracted from the "CLS" token position, and a linear layer is trained jointly with the BERT model for the sentence classification task.

### B.3    CTL

In the CTL model, the M1 model uses exactly the same setup as the SFT model (Equ. 2), the $C$ dimension is chosen as the $\frac{1}{4}$ of the BERT hidden dimension size (Equ. 3), the output dimension of $\Phi$ is chosen to be the same size of the BERT hidden dimension size, and the number of patches is chosen as 10. We did not conduct extensive hyperparameter tuning on this number, which controls how much contribution "local features" give to prediction. Everything is learned end-to-end.

### B.4    CTL-N

The CTL-N model is very similar to the CTL model we defined, except now we use both $C$ and $\Phi$ to make predictions. Conditioning on $X$ introduces a new spurious path between $\sigma$ and $Y$ due to conditioning of the $\Phi$ and $R^1$ colliders, while $S^1$ is unobserved, resulting in the expected drop in OOD performance.

### B.5    CTL-C

In the CTL-C model, only $\mathbf{C}$ is used to predict the outcome $Y$. We observed that CTL-C is a strong alternative predictor, though there may be other unobserved paths influencing $Y$. This is why we introduced $\Phi$ to enable the front-door adjustment.

### B.6    CTL-$\Phi$

CTL-C uses $\Phi$ only to predict the outcome $Y$. We observe that $\Phi$ here captures spurious information.

## C    FURTHER RESULTS

In this section, we first present results of the Yelp semi-synthetic example. We observed a similar trend as Fig. 2

Next, we present an analysis of the impact of the level of spurious information, based on the Amazon semi-synthetic example. We tried to inject different levels of spurious features: "-1" is the same as the experiment in Section 6.1; "-2" means we double the proportion of spurious features, i.e. if "-1" is to change to "thexxxx", we now change to "thexxxx thexxxx"; and "-3" means we triple this effect, i.e. we inject "thexxxx thexxxx thexxxx". We observe that the CTL method consistently outperforms the SFT method under various of spurious information levels.

We also analyze the impact of the training dataset size. While the CTL method consistently outperforms the SFT method, we notice that, as the dataset size increases, the performance gap between CTL and SFT narrows. Specifically, the difference becomes insignificant when approaching 7,000

---

[5]https://github.com/huggingface/transformers

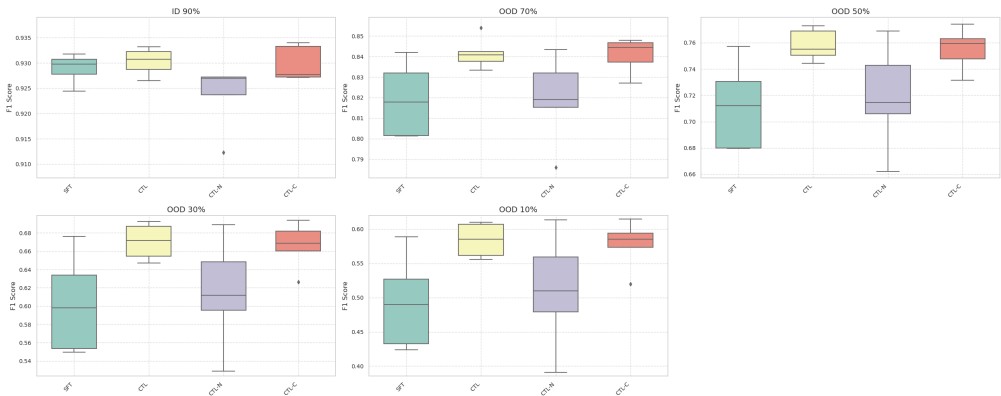

Figure 4: Box-plot over 5 runs for 4 methods (SFT, CTL, CTL-N and CTL-C). Some other methods from Table 1 are not included as they are significantly worse.

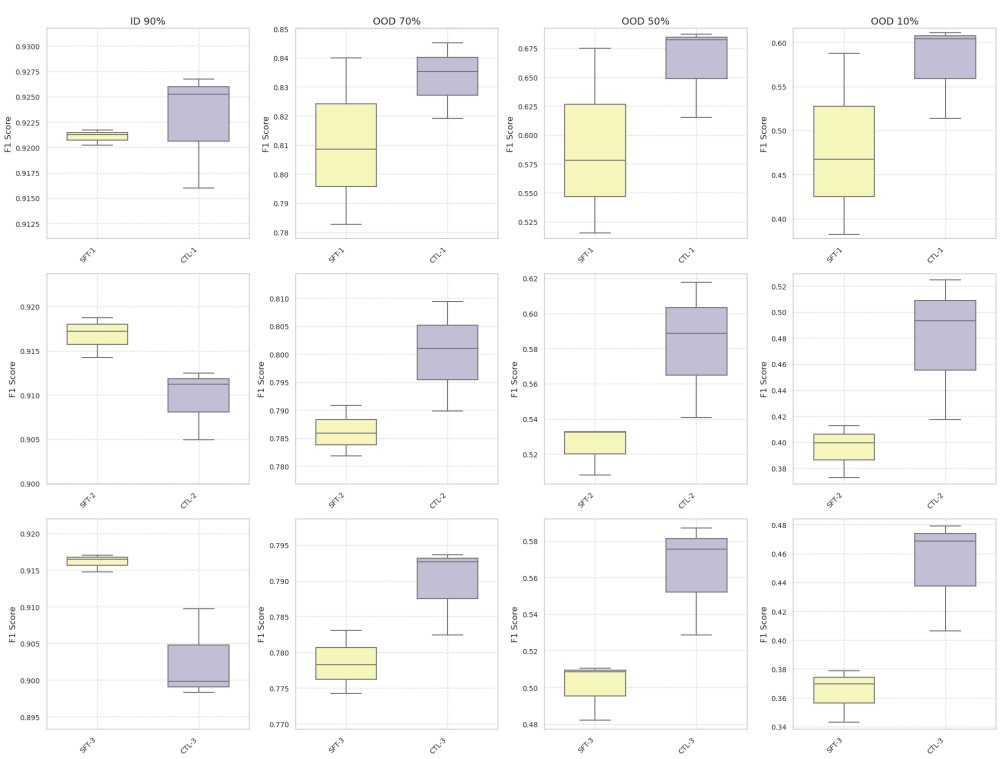

Figure 5: Different spurious level based on the semi-synthetic Amazon data, from "-1" (similarly to the setting in Section 6.1) to "-2" and "-3" with strong spurious features, the CTL consistently outperforms SFT in the OOD settings.

data points per class using the BERT model in our experimental setup described in Section 6.1. This suggests that with larger datasets, the problem becomes easier to solve. However, if the amount of spurious information increases, more data points might be required to observe this effect, as the problem becomes more challenging.

Furthermore, we analyse the impact of the number of $\Phi$ samples used to adjust the causal effect. We can observe from the CTL-N results in Table 1 and 2 that, if we do not adjust for $\Phi$, we get worse results. Also, we observe that that failing to adjust for $\Phi$ leads to worse outcomes. Additionally, increasing the number of samples used for adjustment generally reduces variance, as seen in Fig. 7.

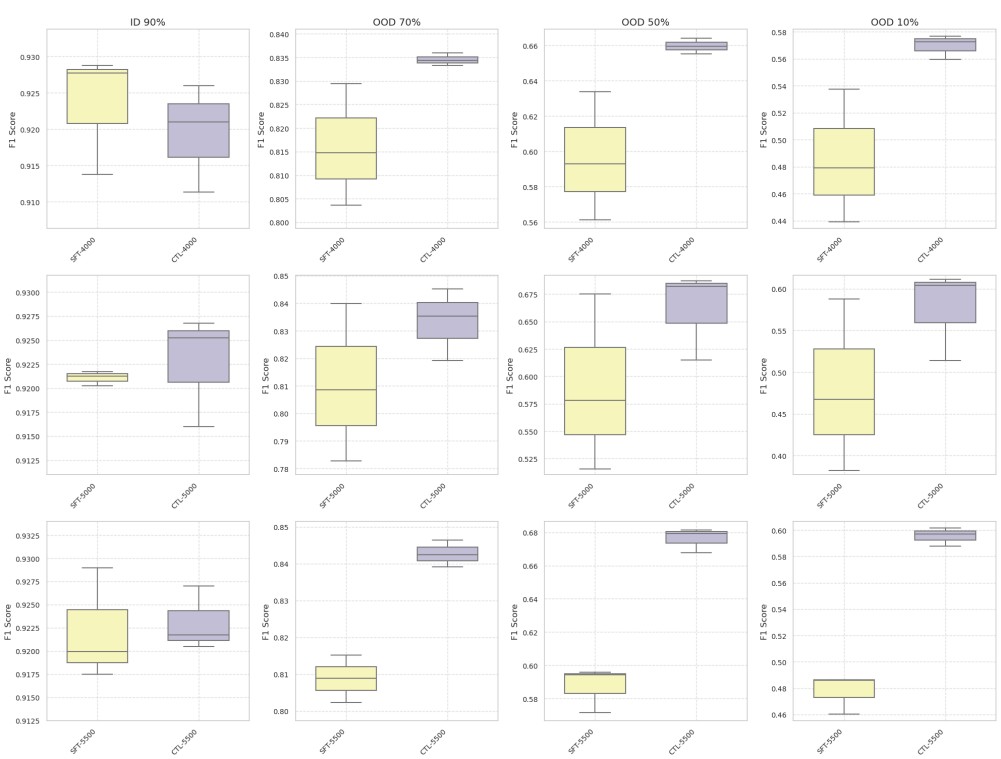

Figure 6: Different training data sizes of 4000, 5000 and 5500 per class of the binary sentiment analysis tasks. The CTL method consistently outperforms SFT in OOD settings.

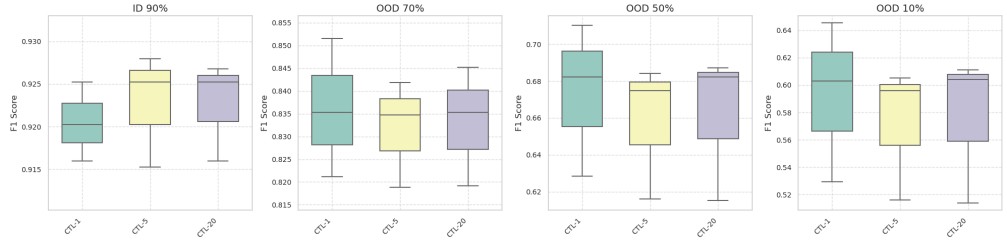

Figure 7: Different inference samples of 1, 5 and 20 for CTL. The variance is reduced in the OOD scenario when using more than 1 sample.

