# OpenReview forum: "Fine-Tuning Pre-trained Language Models for Robust Causal Representation Learning"
_ICLR.cc/2025/Conference — Submitted to ICLR 2025_

### Official Review · Reviewer_AqXT · 2024-10-30

**Soundness:** 3
**Presentation:** 1
**Contribution:** 3
**Rating:** 6
**Confidence:** 3

**Summary:**

The paper proposes to combine a pre-trained language model and a second fine-tuned model to learn two representations of a same input X and use the difference between the two representations to identify the causal factors that remain invariant to the distribution shift induced by fine-tuning. The resulting model is used to compute interventional queries, robust to further distribution shifts. The model is tested on semi-synthetic and real-world sentiment analysis benchmarks in i.i.d and o.o.d settings and outperforms self-supervised baselines.

**Strengths:**

The idea is novel and interesting and tackles the challenging problem of domain generalization. I find the combination between PLMs and causal inference particularly promising. The experiments are convincing as the proposed model performs significantly better in the o.o.d settings.

**Weaknesses:**

Although the idea is interesting, the main issue of the paper is that the theoretical basis is not explained in a clear and sound way. Many typographical errors and undefined terms greatly hinder the comprehension of the paper:

1. The transportability concept, mentioned on line 187, is not properly defined and no references to prior work (e.g. [1,2]) are provided
2. Theorem 2 identifies an interventional query $P(Y|do(X)$ based on the causal graph in Figure 1(c), but the variable X is not in that graph
3. In Equation 3, $r_0$ and $r_1$ are not defined
4. There is an inconsistent use of bold and calligraph fonts, for instance: are $\mathbf{X}$ (appears in Assumption 1), $X$ (appears in Figure 1) and $\mathcal{X}$ (appears in the "Motivations and Intuition" paragraph) the same quantities?
5. In Algorithm 1, line 8, the term $\bar{r_i}$ is used, I assume it should be $\bar{x_i}$
6. Again, in Algorithm 1, line 10, $f_\phi$ is used without being defined
7. I have also found many writing errors (e.g. missing words in sentences or wrongly spelled)


[1] "A General Algorithm for Deciding Transportability of Experimental Results" [Bareinboim and Pearl, 2013]

[2] "Transportable Representations for Domain Generalization" [Jalaldoust and Bareinboim, 2024]

**Questions:**

1. Can you explain further the rationale behind the use of $\tilde{x}$ instead pf $x$ in Equation 2?
2. What is the computational overhead induced by your method compared to other baselines?
3. Have you verified if the proposed method scales to larger PLMs (e.g. Mistral or LLaMA models)?
4. In the analysis of Proposition 1, on line 184,  how does $\sigma$ appear in the expression?
5. In Figure 1(b), shouldn't the edge between $\sigma$ and $X$ be removed too?

---

### Official Review · Reviewer_3XZN · 2024-11-05

**Soundness:** 3
**Presentation:** 3
**Contribution:** 3
**Rating:** 5
**Confidence:** 2

**Summary:**

This paper focuses on the out-of-distribution (OOD) generalization of language models (LMs). The authors hypothesize that the poor OOD generalization of LMs stems from their reliance on non-causal representations. To address this issue, the authors propose a causal front-door adjustment to augment the training data. Empirical evaluations on small-scale LMs demonstrate the effectiveness of the proposed method.

**Strengths:**

The motivation behind this paper and the proposed method is reasonable and well-justified.

**Weaknesses:**

**Unclear Method Description.** The authors use extensive mathematical derivations to analyze and explain the OOD generalization of LMs from a causal inference perspective. Although I appreciate the authors' efforts, I find it unclear how the method is actually implemented, particularly in Section 5. I would appreciate it if the authors could detail the implementation process without relying on math, specifically how a local value is constructed (Section 5.3) and how the model is trained and used for inference (Section 5.4).

**(Possibly) Insufficient Evaluation.** While the proposed method shows effectiveness in the experiments, the choice of datasets appears relatively simple, and the LM backbone is somewhat limited. The authors use two sentiment classification datasets for evaluation, with `bert-base-uncased` as the LM backbone.

- Regarding the model, given the significant advancements in existing large language models (e.g., LLaMA3-8b), I am concerned that the proposed method may not remain effective, as such models can easily handle a simple task like sentiment classification and achieve near-perfect performance. It would be helpful if the authors could demonstrate the method's effectiveness on stronger models.
- Regarding the task, I also recommend that the authors evaluate their method on more challenging and commonly-used datasets, such as instruction-following and safety alignment. For example, some large language models like LLaMA-2-7B-chat are known to be over-conservative. During training (RLHF), they are fine-tuned to reject answering user questions that may lead to unsafe outputs. However, this often results in over-conservatism, causing the model to reject many safe requests. Could this be a sign of poor OOD generalization (in terms of safety) in these models? Is it possible to apply the proposed method to these tasks?

I acknowledge that the second suggestion may be challenging to implement, but I encourage the authors to consider tasks beyond simple sentiment classification.

**Writing Issues:** There are several writing issues in the paper:

- What do $R_1$ and $R_0$ mean? When the authors introduced them in Line 243, they refer to "variations of its (text's) representations," but in Line 261, they claim "r getting its sentence summary $R_1$ and $R_0$."
- Potential typos: Line 336: "obtrain" should be "obtain."
- Possible errors in Table 2: SWA outperforms the proposed method in the 2nd and 3rd columns.
- Some full names and abbreviations are redundant, such as "pretrained language models (PLMs)" in the conclusion section. Since PLM was already defined earlier, it is unnecessary to restate the full term. If this section was revised by ChatGPT or another LLM (which often introduces abbreviations in its revisions as it does not have the full context of previous sections), I recommend that the authors review and remove these repetitions.

However, since I am not an expert in causal inference, I acknowledge that my assessment may be limited. I will adjust my review based on other reviewers' feedback.

**Questions:**

Please refer to the weakness above.

---

### Official Review · Reviewer_8Kzk · 2024-11-08

**Soundness:** 1
**Presentation:** 1
**Contribution:** 2
**Rating:** 5
**Confidence:** 2

**Summary:**

This paper proposes to solve OOD problems for LLM prediction tasks through causal invariance learning. It seeks to learn the causal feature through paired representations of the same input but under different models (pre-trained vs SFT). Then it produces p(y|do(x)) with the causal features.

**Strengths:**

The paper studies an important question: how to mitigate OOD in LLM prediction.

**Weaknesses:**

* I find the paper difficult to follow
* The developed method is overly complicated, and there might be simpler ways to do it
* The assumptions are very strong, but meanwhile I don't see the clear motivation of some design choices
* There are many typos, sometimes in importance places (e.g. Step 4 in Algo 1)
I will elaborate in questions part. I can adjust the points if my questions are addressed.

**Questions:**

* One important task is to identify the causal feature C (e.g. in the sentiment classification task, it is the true sentiment concept, as opposed to the spurious feature such as the platform). There are many many methods for estimating it when you have samples from diverse environment. A natural choice of environment seems to be the different platforms, say. But this paper uses the representation of the same input from the two "environment": one from pre-trained checkpoint, and the other from SFT checkpoint. I don't understand the motivation, and I don't understand how the assumption of Thm 2 (Thm 4.4 of https://arxiv.org/pdf/2106.04619) is satisfied (since we don't generate x under these models? ) I hope the authors can elaborate on this.
* If we already learned C, and in assumption 1 we assume y ~ p(y|c), why don't we directly predict Y based on C? I may be missing something so I hope the authors can help me understand.
* Equation 1 seems very complicated and the estimation could easily compound. What can we do about it?

---

### Official Review · Reviewer_k9Gi · 2024-11-11

**Soundness:** 3
**Presentation:** 3
**Contribution:** 3
**Rating:** 6
**Confidence:** 3

**Summary:**

The paper introduces a causal model to combat spurious correlations in NLU tasks and provides an algorithm to learn its causal features. Experimental results are provided at various strengths of spurious correlations, demonstrating their algorithm's generalization capability on OOD test sets where the traditional i.i.d. assumption is violated.

**Strengths:**

- The use of the front-door adjustment at the embedding level for LLMs appears to be novel. Recent work (e.g., https://arxiv.org/abs/2403.02738) applied it at the prompt level.
- Assumptions are clearly stated and experimental results are provided across different instantiations of authors' method.
- Methodology built on established self-supervised causal representation work in https://arxiv.org/pdf/2106.04619.

**Weaknesses:**

- The semi-synthetic and real-world experiments appear very similar. Both appear to be based on Amazon and Yelp review datasets, but the injected text of the real-world experiments are based on data source. Exploring datasets suggested in https://arxiv.org/pdf/2311.08648 may strengthen the experiments section.
- More details on the connection to Theorem 4.4 in https://arxiv.org/pdf/2106.04619 would be appreciated.

**Questions:**

- In Table 2, it appears the SWA method does quite well even compared to CTL. Is there a good explanation for this? Furthermore, SWA and WISE are excluded from Table 1 so it is hard to tell if CTL is empirically better. Additionally, in Table 2, the 99.99% Train F1 of SWA is not bolded.
- Inference for an individual x_i depends on the mini-batch it is within. How much variance is there in predictions across batch sizes?

---

### Official Review · Reviewer_WfKd · 2024-11-12

**Soundness:** 2
**Presentation:** 2
**Contribution:** 3
**Rating:** 6
**Confidence:** 3

**Summary:**

The authors focused on causal representation learning in NLP applications for robust generalization on OOD data. The conventional method for OOD generation is leveraging data from multiple domains as a feature augmentation, which has limitations in the natural language domain in that multi-domain data is often not readily available nor straightforwardly applicable. The authors proposed leveraging PLMs as an additional source of domain data for efficient data augmentation based on the causal front door adjustment to reduce spurious correlation in data. By evaluating text classification tasks, from single-domain scenarios under mild assumption to targeted more general real-world scenarios, the authors demonstrated the generalizability of LLM improved.

**Strengths:**

- The paper is well written, with a kind introduction of causality to NLP application (text classification)
- The authors provide the proper theoretical formulation to apply causality for future NLP application research. The problem formulation to address robust generalization to OOD is a more principled way to tackle OOD generalization than domain adaptation based or attention based method; Estimate prediction on shifted test data $P*(y|do(x))$ by prediction on in-domain training data $P*(y|do(x))$, with assumptions in Sec 4.1.

**Weaknesses:**

- In Sec 4.2 theorem 1, the authors assume that the causal latent variable C between two environments (pre-trained and fine-tuned models) stays the same. However, this assumption seems a kind of logical leap, as there is no supporting experimental analysis or general consensus, and at the same time, practically not trivial. In other words, this assumption needs that practical solution (good fine-tuned model) is already fulfilled; if there exists a causally robust language model on each downstream task, why do we need OOD generalization of the pre-trained model?
- Experiments are monotonic, so it is hard to comprehensively evaluate whether the suggested method increases OOD generalization.
    - The available experimental results in the main body seem to repeat the same experimental design over just two datasets (though there are plenty of available classification benchmarks), which is insufficient for holistic understanding. Could you provide additional experiments on other dataset in diverse domain?
    - There is no experimental analysis other than performance, so it is hard to understand how the designed algorithm actually works regarding causality. Could you provide more in-depth analysis not only performance?

**Questions:**

- Extending this work to generative LLM would be interesting and, at the same time, of practical importance. How can it be possible?
- In fig (c), it seems weird that the token embedding $\Phi$ is the output of language model given input $R^1 \sim p(r_1(x)|s_1, c)$ again affects causal latent variable c, because the embedding is the result of c. In the NLP application, how can we understand the causal order of the cyclic relation R^1 \to \Phi C \to R^1$? Is it an iterative loop?
- Sec 5.1 eq(2)
    - The authors state that learning $(\tilde{x}, y)$ improved the performance stability, but what makes the difference between $x, \tilde{x}$?
    - The authors said $l$ is the cross-entropy loss, but the function argument is $f(\tilde{x})-y$, which is at most single probability distribution. How did you calculate the cross-entropy of which two probability distributions?

---

### Meta-Review · Area_Chair_dkqK · 2024-12-21

**Metareview:**

This paper studies domain generalization behavior in supervised learning in the setting where the predictor is a pre-trained language model. Nominally, the goal is to learn the distribution P(Y | do(x)), on the basis that this will have some robustness to spurious correlations.

This paper generated considerable disagreement among the reviewers, so I read it myself. Ultimately, I agree with the reviewer criticisms that found the paper unclear in its exposition and assumptions (and thus also in its applicability). In particular, I found the explanation of the methodology difficult to follow, and had an even harder time understanding the connection between the theory and the method. These confusions seem to be common among the reviewers, though they vary in how much they are troubled by this. It is certainly possible that this is a good piece of work, but it is also clear that there at least needs to be a significant reworking for clarity before it can have real impact. As such, my view is that it is not yet ready for publication.

Beyond the issues of clarity, I have to confess that I'm a bit worried about whether domain shifts in supervised prediction from text is still a real problem. Particularly, the dominant approach has shifted away from ERM to instead simply framing the prediction task in natural language (possibly with few shot examples) and using an LLM to complete it. It would be good to have at least one example of a problem where this more modern approach doesn't simply solve the language prediction problem (thus wholly circumventing the question of domain shifts)

**Additional Comments On Reviewer Discussion:**

See above

---

### Decision · Program_Chairs · 2025-01-22

Reject